# Optimizing the Retrieval of Wheat Crop Traits from UAV-Borne Hyperspectral Image with Radiative Transfer Modelling Using Gaussian Process Regression

**Rabi N. Sahoo [1,*], Shalini Gakhar [1], Rajan G. Rejith [1], Jochem Verrelst [2], Rajeev Ranjan [1], Tarun Kondraju [1], Mahesh C. Meena [3], Joydeep Mukherjee [1], Anchal Daas [4], Sudhir Kumar [5], Mahesh Kumar [5], Raju Dhandapani [5] and Viswanathan Chinnusamy [5]**

[1]  Division of Agricultural Physics, Indian Council of Agricultural Research (ICAR)—Indian Agricultural Research Institute (IARI), Pusa, New Delhi 110012, India; shalinigakhar7@gmail.com (S.G.); rejithrg01@gmail.com (R.G.R.); rajeev.ranjan@icar.gov.in (R.R.); tarunkondraju@gmail.com (T.K.); joydeep.mukherjee@icar.gov.in (J.M.)
[2]  Image Processing Laboratory (IPL), Parc Científic, Universitat de València, 46980 Paterna, Spain; jochem.verrelst@uv.es
[3]  Division of Soil Science & Agricultural Chemistry, Indian Council of Agricultural Research (ICAR)—Indian Agricultural Research Institute (IARI), Pusa, New Delhi 110012, India; mahesh.meena@icar.gov.in
[4]  Division of Agronomy, Indian Council of Agricultural Research (ICAR)—Indian Agricultural Research Institute (IARI), Pusa, New Delhi 110012, India; anchal.dass@iari.res.in
[5]  Division of Plant Physiology, Indian Council of Agricultural Research (ICAR)—Indian Agricultural Research Institute (IARI), Pusa, New Delhi 110012, India; sudhir.kumar4@icar.gov.in (S.K.); r.dhandapani@icar.gov.in (R.D.); v.chinnusamy@icar.gov.in (V.C.)
*   Correspondence: rabi.sahoo@icar.gov.in

**Abstract:** The advent of high-spatial-resolution hyperspectral imagery from unmanned aerial vehicles (UAVs) made a breakthrough in the detailed retrieval of crop traits for precision crop-growth monitoring systems. Here, a hybrid approach of radiative transfer modelling combined with a machine learning (ML) algorithm is proposed for the retrieval of the leaf area index (LAI) and canopy chlorophyll content (CCC) of wheat cropland at the experimental farms of ICAR-Indian Agricultural Research Institute (IARI), New Delhi, India. A hyperspectral image captured from a UAV platform with spatial resolution of 4 cm and 269 spectral bands ranging from 400 to 1000 nm was processed for the retrieval of the LAI and CCC of wheat cropland. The radiative transfer model PROSAIL was used for simulating spectral data, and eight machine learning algorithms were evaluated for hybrid model development. The ML Gaussian process regression (GPR) algorithm was selected for the retrieval of crop traits due to its superior accuracy and lower associated uncertainty. Simulated spectra were sampled for training GPR models for LAI and CCC retrieval using dimensionality reduction and active learning techniques. LAI and CCC biophysical maps were generated from pre-processed hyperspectral data using trained GPR models and validated against in situ measurements, yielding $R^2$ values of 0.889 and 0.656, suggesting high retrieval accuracy. The normalised root mean square error (NRMSE) values reported for LAI and CCC retrieval are 8.579% and 14.842%, respectively. The study concludes with the development of optimized GPR models tailored for UAV-borne hyperspectral data for the near-real-time retrieval of wheat traits. This workflow can be upscaled to farmers' fields, facilitating efficient crop monitoring and management.

**Keywords:** machine learning (ML) Gaussian process regression (GPR); unmanned aerial vehicle (UAV); hyperspectral; ARTMO; wheat; radiative transfer model

## 1. Introduction

Biophysical variables such as leaf area index (LAI), leaf chlorophyll content (LCC), and canopy chlorophyll content (CCC) are the key traits to monitor crop growth [1]. LAI is

defined as half the total leaf area per unit horizontal ground surface area [2], is strongly associated with photosynthetic production, respiration, and transpiration properties of plants, and is one of the key factors for crop yield [3]. The efficient estimation of LAI is an indispensable process to ensure steadfast support towards optimised field management practices for target yield [4]. LAI can be either directly or indirectly estimated. Direct methods involve the completely destructive way of leaf sampling and then measuring the LAI using a portable leaf area meter. Indirect methods involve Beer–Lambert law-based optical methods and inclined point quadrat methods [5]. Here, non-destructive optical instruments such as the LAI-2000 plant canopy analyser (Li-Cor, Inc., Lincoln, NE, USA) measure the radiation from the canopy and accurately infer LAI data [6]. Nowadays, even smartphone apps have been developed for indirectly measuring LAI in the field [7,8]. In fact, there has been a long history of remote LAI estimation using various imaging approaches, such as directional photography, multispectral photography, laser scanning, etc., as reported by [5]. Specifically, in the optical remote sensing domain, various LAI retrieval techniques, such as vegetation indices (VIs), machine learning regression algorithms, inversion of canopy radiative transfer models (RTMs), and recently hybrid models, have also been deployed using multispectral and hyperspectral data [9–13].

The chlorophyll pigment provides green colour to the plant and determines the photosynthetic rate along with its productivity [14]. LCC can be considered an imperative indicator to analyse crop growth and development [15]. The dynamics of LCC depend upon a multitude of seasonal and environmental factors, such as intensity of light, temperature, and so on [16]. Plant chlorophyll content is often estimated based on the reflectance, transmittance, and absorbance of significant wavelengths of the electromagnetic spectrum that interact with leaves, either by using hand-held chlorophyll meters or chemicals (such as dimethyl sulfoxide) [17–19]. Further, CCC is the total amount of chlorophyll a and b pigments existing in the photosystems of leaves in a contiguous group of plants per unit of ground area [20]. CCC is computed as the product of LAI and LCC per unit leaf area [21]. Numerous spectral indices have been proposed for the assessment of CCC (e.g., [22–24]). Yet, the selection of optimal retrieval models remains challenging, because their performance is highly dependent upon the hyperparameters, targeted plant species, ground sampling, hyperspectral data, and various intricate factors [24–26].

Lately, a comprehensive analysis of diverse hyperspectral datasets across different species and geolocations has been carried out due to informative narrow spectral bands over a continuous range of the electromagnetic spectrum [27–29]. Unlike destructive techniques for the estimation of biochemical traits, hyperspectral imaging is a non-destructive, non-invasive, near-real-time, efficient, and robust method. This technology proved to be promising for plant phenotyping that incorporates indoor (controlled environment) and outdoor (field-scale) phenotyping [30]. Hyperspectral sensors mounted on unmanned aerial vehicles (UAVs) were deployed for imaging on large farms [31–33]. Flexible, low-cost, and high-resolution optical sensor systems mounted on UAV platforms are important for bridging data gaps and supplementing the capabilities of manned aircraft and satellite remote sensing systems [34].

The ultrahigh-spectral- and -spatial-resolution data obtained with UAVs amalgamated with various in-field biophysical and biochemical variables have opened opportunities to remotely quantify unexplored traits of plants [35]. Further, this necessitates the evolution of data processing and analysis using machine learning algorithms for the estimation of multiple vegetation traits. These techniques are appealing for processing hyperspectral data due to their flexible nonparametric behaviour and the excavation of unknown facts from raw data [26]. They readily explore the nonlinear relationship between the spectral reflectance values and target crop biophysical variables like LAI, LCC, and CCC.

In all generality, these variables may be retrieved from hyperspectral data using (i) parametric regression, (ii) nonparametric regression, (iii) inversion of RTMs, and (iv) hybrid or combined methods [26,36]. In the case of parametric regression, a limited number of spectral channels with significant information are commonly used to mathematically

formulate a fitting function (linear or nonlinear) between the selected channels and crop traits. Unlike the parametric approach, nonparametric methods make use of the full spectral range and include a learning stage for model training. They often include the optimisation of hyperparameters, which govern the efficiency of a model. Stepwise multiple linear regression (SMLR), partial least squares regression (PLSR), decision trees, artificial neural networks (ANNs) [37,38], support vector regression (SVR) [39], genetic algorithms [40], and Gaussian process regression (GPR) [41,42] are some of the standard machine learning techniques used for crop variable estimation in this category.

Regarding RTM inversion, cause–effect relationships are exploited by inverting the RTM against actual spectral signatures to retrieve crop traits. However, the number of unknowns is greater than the independent variables, making the model hard to invert [43]. Finally, hybrid models combine the capabilities of physical methods such as RTMs with nonparametric regression methods. PROSPECT-4 is extensively used to simulate leaf optical features [44], and SAIL is for solving the scattering and absorption equations at the canopy level [45,46]. All possible combinations of input variables may be formed for multiple model realizations to store in the directory called look-up table (LUT). Additionally, several experimental studies demonstrated the superiority of an emerging Bayesian nonparametric modelling approach known as GPR as opposed to alternative machine learning algorithms in the accurate retrieval of different crop variables [47–50]. GPR is similar to kernel ridge regression and kriging, with the additional provision of band relevance ranking [11] and mapping of associated uncertainty estimates [51]. Among others, GPR was used to estimate the forest's LAI using airborne hyperspectral data in the range of 400 to 2500 nm [42].

Even though several studies presented hybrid models for estimating crop traits from hyperspectral remote sensing datasets (e.g., [11,52–57]), UAV datasets have not yet been fully explored. A few studies focused on standard LUT inversion [45,58–60] and the particle swarm optimization algorithm [61] applied to PROSAIL simulations for retrieving phenotypic information of crops from UAV hyperspectral data. Recently, hybrid retrieval of crop traits applied to UAV-based VNIR hyperspectral data has been explored [58]. Yet, the advantages of using active learning (AL) methods, where the training of the model is optimized based on only the most relevant samples, are still to be considered. In hybrid retrieval approaches, AL techniques provide an intelligent approach to informative sampling from large data pools. Concerning the development of hybrid GPR models, AL techniques are used to minimize a large set of PROSAIL simulations by only selecting samples that contribute to improving model accuracy. By developing such light models, they speed up the runtime [55].

Altogether, this study is dedicated to developing, optimizing, and validating hybrid GPR models for retrieving the wheat biophysical traits of LAI and CCC using an ultrahigh-resolution UAV hyperspectral dataset. Specifically, this study aims to (i) investigate the performance of hybrid non-kernel and kernel machine learning regression algorithms for retrieving wheat crop traits from UAV hyperspectral data; (ii) optimize a hybrid GPR retrieval model using suitable AL sampling methods; and finally, (iii) validate the retrieved maps against in situ measurements and mapping.

## 2. Materials and Methods

The block diagram of the methodology used for the proposed approach is shown in Figure 1. The main steps involved in the workflow are (i) field experimentation, UAV image acquisition, and pre-processing; (ii) PROSAIL simulations and model evaluation; (iii) Gaussian process regression (GPR); (iv) dimensionality reduction using principal component analysis (PCA); and (v) active learning methods and field verification. Each step is explained in detail in the subsequent sections.

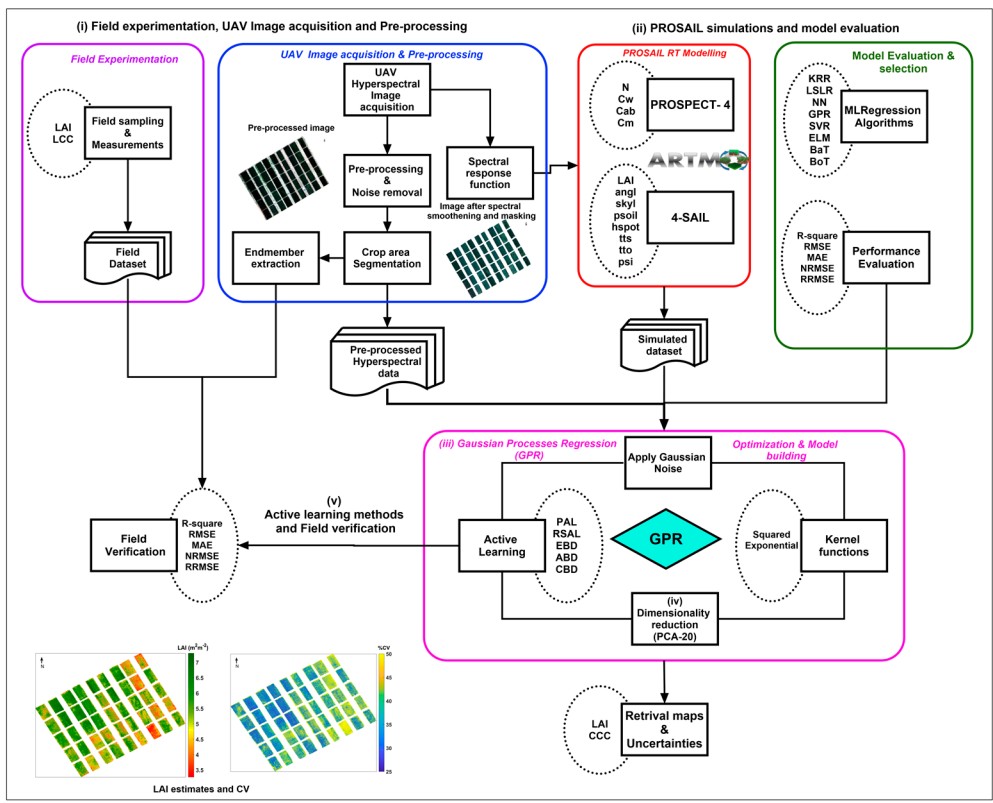

**Figure 1.** Block diagram of the methodology used for the study.

### 2.1. Field Experimentation and UAV Image Acquisition

This study targeted an experimental wheat field of the HD 3059 variety located at the research farm of Indian Council of Agricultural Research-Indian Agricultural Research Institute (ICAR-IARI) (28°38′28.314″N latitude and 77°9′3.106″E longitude) 228 m above mean sea level. The wheat field consisted of three replications of fifteen plots (7.2 × 13 m size each), maintained at five graded nitrogen levels (0, 50, 100, 150, and 200 kg N ha$^{-1}$), and three irrigation treatments (soil moisture sensor-based treatment (I$_1$); crop water stress index (CWSI)-based treatment (I$_2$); and conventional treatment (I$_3$)). The study area map showing the location of experimental plots and wheat fields is shown in Figure 2. The experiment was carried out during the winter season of 2021–2022, with the crop being sown on 13 December 2021, and harvested on 15 April 2022. LCC was estimated using the dimethyl sulfoxide (DMSO) method [62], and LAI was measured using an LAI-2000 plant canopy analyser (Li-Cor, Inc., Lincoln, NE, USA) [63]. Three measurements were taken from each plot, and their average was used for the analysis. Finally, CCC was obtained by multiplying LCC with LAI (LAI × LCC) [64].

The experimental plots were overflown by a Headwall Nano-Hyperspec hyperspectral camera (Headwall Photonics Inc., Bolton, MA, USA) mounted on a UAV hexacopter on 17 March 2022. The hyperspectral image composed of 269 bands in the range of 400–1000 nm with a spectral interval of 2.2 nm and spatial resolution of 4 cm was captured at a flight height of 21 m. UgCS Mission planning software was used for planning the mission route. Headwall SpectralView (v3.1.4) software (Headwall Photonics, Bolton, MA, USA) and ENVI (version 5.6.3) were employed for processing the acquired hypercube. After image acquisition, the pre-processing steps of radiance correction, reflectance conversion, orthorectification, and image mosaicking were accomplished using the aforesaid software. Subsequently, spectral noise was removed using the Savitzky–Golay filter [65,66]. The wheat cropland area was segmented using a binary mask generated by multiplying the hyperspectral normalized difference vegetation index (hNDVI) threshold and the spectral angle mapper (SAM) classified map [67]. The endmember spectra corresponding to in situ

crop traits were generated using a sequential maximum angle convex cone (SMACC) [68]. Compared with the pixel purity index (PPI), the SMACC shows better performance in extracting endmembers of vegetation from hyperspectral imagery [69]. During endmember collection, an inner buffer distance of 1 m from the borders was excluded, and a mean spectrum of thirty endmembers from each plot was taken to attain accuracy. These trait-specific in situ training datasets were used for optimization and validation.

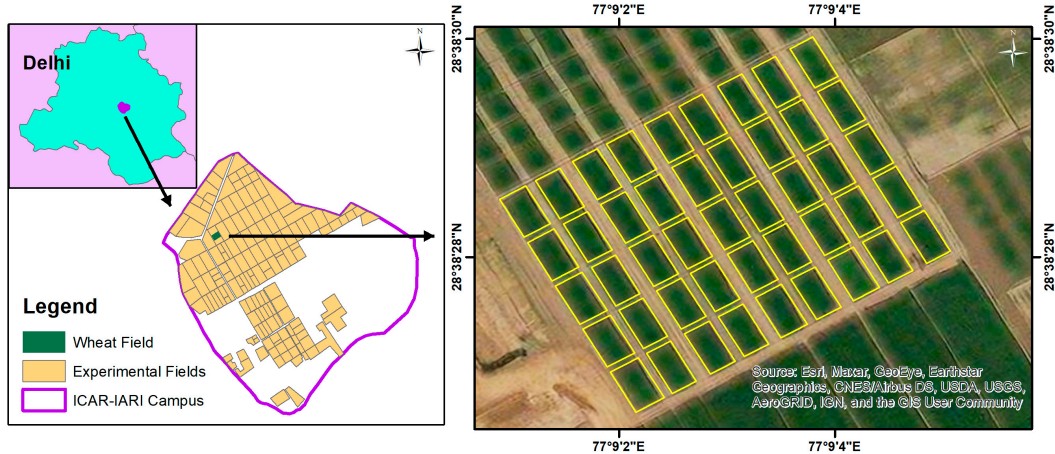

**Figure 2.** Location map of study area showing the experimental wheat fields at the research farm of ICAR-IARI, New Delhi.

### 2.2. PROSAIL Simulations and Model Evaluation

The simulated dataset from PROSAIL RTM was used for retrieving wheat biophysical traits. PROSAIL is a combined RTM of the PROSPECT-4 leaf reflectance model and the 4-SAIL canopy reflectance model [70]. The PROSPECT-4 model consists of four variables, leaf structure coefficient ($N$), leaf chlorophyll content ($C_{ab}$), equivalent water thickness ($C_w$), and dry matter content ($C_m$). The PROSPECT model generates simulated directional reflectance in the range of 400 to 1000 nm at the spectral resolution of 2.2 nm. The output was used by the SAIL model for generating hemispherical and bi-directional top-of-canopy (TOC) reflectance, in which the bi-directional TOC reflectance is used as the basis for retrieving the wheat biophysical variables. The input variables for 4-SAIL are LAI, average leaf angle ($angl$), fraction of diffuse incoming solar radiation ($skyl$), soil brightness coefficient ($psoil$), hot-spot size parameter ($hspot$), solar zenith angle ($tts$), sensor zenith angle ($tto$), and relative azimuth ($psi$). The sampling range for each parameter of the PROSPECT-4 and 4-SAIL models (Table 1) was selected from field measurements and previous studies carried out at IARI, New Delhi, in wheat fields [37,71–74], as well as similar studies carried out in other wheat fields using RT modelling [56]. The sampling size of the LUT was fixed to 2000 simulations using Latin hypercube sampling (LHS) across the parameter space. LHS is considered to be an adequate sampling size for hyperspectral imagery to enable fast processing with accurate results [56]. In hybrid GPR approaches, low sampling sizes of a few hundred were used for developing light models with low runtime. The sampling size reported for various GPR retrieval models for space-borne hyperspectral datasets ranges from 1000 to 2000 [53,56,75].

The ARTMO (Automated Radiative Transfer Models Operator) software package is a modular MATLAB-based toolbox embedded with, among others, multiple leaf and canopy RTMs, and machine learning regression algorithm (MLRA) toolboxes, which facilitate automated model analysis and vegetation mapping [48,76]. The ARTMO toolbox was successfully employed for generating the PROSAIL simulations by combining PROSPECT-4 with the SAIL model [70]. Seven regression algorithms, as listed in Table 2, and GPR were evaluated for finding the most accurate model. The prediction accuracy was measured using the standard goodness-of-fit statistical parameters of mean absolute error (MAE),

root mean square error (RMSE), relative RMSE (RRMSE), normalized RMSE (NRMSE), and coefficient of determination ($R^2$) with the help of 70% training data and the remaining 30% validation data. The entire processing took place in the computer system with specifications of Windows-64 OS, Intel(R) Xeon(R) W-2133 CPU @ 3.60 GHz, 32.0 GB RAM.

**Table 1.** List of input variables of the PROSAIL model used in the study with variable range.

| S. No. | Parameter | Abbreviation | Unit | Values |
|---|---|---|---|---|
| | | Leaf Model: PROSPECT-4 | | |
| 1. | Leaf structure coefficient | N | Dimensionless | 1 |
| 2. | Equivalent water thickness | $C_w$ | cm | 0.01–0.045 (0.001 interval) |
| 3. | Leaf chlorophyll content | $C_{ab}$ | $\mu cm^{-2}$ | 0–80 (0.2 interval) |
| 4. | Dry matter content | $C_m$ | $g\,cm^{-2}$ | 0.0046 |
| | | Canopy Model: 4-SAIL | | |
| 5. | Leaf area index | LAI | $m^2\,m^{-2}$ | 0.1–7.5 (0.01 interval) |
| 6. | Average leaf angle | angl | Degree | 70, 57, 45 |
| 7. | Fraction of diffuse incoming solar radiation | skyl | Dimensionless | 0.1 |
| 8. | Soil brightness coefficient | psoil | Dimensionless | 0.1 |
| 9. | Hot-spot size parameter | hspot | $mm^{-1}$ | 0.78, 0.40, 0.32 |
| 10. | Solar zenith angle | tts | Degree | 51, 45, 33 |
| 11. | Sensor/view zenith angle | tto | Degree | 0 |
| 12. | Relative azimuth | psi | Degree | 0 |

**Table 2.** List of evaluated regression algorithms.

| S. No. | Algorithm | Principle | Formula and Description |
|---|---|---|---|
| 1. | Kernel ridge regression (KRR) [77] | - Combines ridge regression and classification along with kernel trick.<br>- Uses squared error loss.<br>- Faster for medium-sized datasets. | $y = w^T \Phi(x) = y\left(\Phi^T \Phi + \lambda I_n\right)^{-1} \Phi^T \Phi(x) = y(K + \lambda I_n)^{-1} \kappa(x)$<br>where $x$ is the new test point, $w$ is the solution, $K(bx_i, bx_j) = \Phi x_i{}^T \Phi(x_j)$, and $K(x) = K(x_i, x)$. Adding bias $\Phi : \Phi_o = 1$.<br>$w^T \Phi = \sum_a w_a \Phi_{ai} + w_o$ |
| 2. | Least squares linear regression (LSLR) | - Approach to model the relationship between a dependent variable and one or more independent variables.<br>- Does not consider the complexity of data. | $m = \frac{\sum_{i=1}^{n}(x_i - \overline{x})(y_i - y)}{\sum_{i=1}^{n}(x_i - \overline{x})^2}$<br>$c = \overline{y} - m\overline{x}$<br>where $\overline{x}$ is the mean of all the values in input $x$ and $\overline{y}$ is the mean of all the values in desired output y. m is the slope of the line, and $c$ is the y-intercept. |
| 3. | Neural network (NN) | - Approach that uses a standard back-propagation algorithm applied to a set of input, hidden, and output layers.<br>- Predicts the results for unknown datasets.<br>- Requires labelled data for the training process.<br>- The training of the network takes time. | $y_i = g_i = g(\sum_{j=1}^{K} w_{ji} x_j + \theta_i)$<br>$E = \frac{1}{2} \sum_{J=1}^{k} (y_j - t_j)$<br>where $y$ —output; $x$ —input; $g$ —activation function; $w$ —weight; $\theta$ —bias; $E$ —error; and $t$ —ground truth, for instance. |

**Table 2.** *Cont.*

| S. No. | Algorithm | Principle | Formula and Description |
|---|---|---|---|
| 4. | Support vector regression (SVR) | - Works on the concept of maximizing the margins.<br>- Generates a decision boundary with maximum separation.<br>- Proves helpful when multiple heterogeneous classes are available. | $width = \frac{1}{2} \cdot \frac{w^2}{\|w\|}$<br><br>$L = \sum_{i}^{n} a_i - \frac{1}{2}\sum_{i}^{n}\sum_{j}^{n} a_i a_j y_i y_j \vec{x_i} \cdot \vec{x_j}$<br><br>$k(x,y) = x^T y + c$<br><br>where $x$ is the sample dataset, for which SVM finds weights $w$ such that the data points in the dataset are separated using the most optimum hyperplane. Further, $L$ is differentiated with respect to $w$. |
| 5. | Extreme learning machine (ELM) | - Learning algorithm for single-layered feed-forward neural network.<br>- Higher accuracy as compared with SVM and neural networks.<br>- Fast learning.<br>- Computationally scalable.<br>- Independent from the tuning process.<br>- Evaluation speed is low.<br>- Requires astronomically high hidden-layer neurons.<br>- Cannot encode more than one layer of abstraction. | $\sum_{i=1}^{L} \beta_I G(a_i, b_i, x), x \in R^d, \beta_I \in R^m$<br><br>$G(a_i, b_i, x) = g(a_i.x + b_i), a_i \in R^d,$<br>$b_i \in R$<br><br>$\sum_{j=1}^{N} \|f_L(x_j) - t_j\| = 0$<br><br>$\sum_{i=1}^{L} \beta_I G(a_i, b_i, x) = t_j, j = 1, 2 \ldots\ldots N$<br><br>$H\beta = T$<br><br>where $L$—hidden nodes; $G(a_i, b_i, x)$—output function at $i$th hidden node; $a_i, b_i$—hidden node parameters; $\beta_i$—weight vector; $g$—activation function; and $H$—hidden-layer output matrix. |
| 6. | Bagging trees (BaTs) | - General-purpose procedure for reducing the variance of a statistical learning method.<br>- Makes predictions on the tree's out-of-bag observations.<br>- Multiple trees can be trained simultaneously.<br>- All the trees trained on different bootstrap samples are correlated. | $\widehat{f_{bag}} = \widehat{f_1}(X) + \widehat{f_2}(X) + \cdots + \widehat{f_b}(X)$<br>where $X$ is the record for which a prediction is to be generated, $\widehat{f_{bag}}$ is the bagged prediction, and $\widehat{f_1}(X) + \widehat{f_2}(X) + \cdots + \widehat{f_b}(X)$ are the predictions from the individual base learners. |
| 7. | Boosting trees (BoTs) | - Transforms weak decision trees (called weak learners) into strong learners.<br>- Tends to overfit.<br>- Better than random predictions.<br>- Good at handling tabular data with numerical features.<br>- Able to capture nonlinear interactions between the features and the target.<br>- Not designed to work with very sparse features. | $g(x) = f_o(x) + f_1(x) + f_2(x) + \cdots$<br>where final classifier $g$ is the sum of simple base classifiers $f_i$.<br><br>$x_{t+1} = x_t - \eta \frac{\delta f}{\delta x}\Big\|_{x=x_t}$<br>where $\eta$ is called the step size.<br><br>$f_t = \arg min \sum_{i=1}^{N} \left[ \frac{\delta L(y_i, g(x_i))}{\delta g(x_i)}\Big\|_{g=g_t} - f(x_i) \right]^2$<br><br>where $t$ is the iteration; $L(y_i, g(x_i))$ is the empirical loss function at each iteration; and $g_t$ is moved towards the negative gradient direction. |

### 2.3. Gaussian Process Regression (GPR)

As opposed to other MLRAs, such as NN and KRR, GPR has proven to be a powerful regression model with two major advantages [48]. First, it provides an additional quantitative measurement of prediction accuracy in terms of uncertainty estimates (σ). A lower σ indicates a better prediction for crop traits [54]. The second advantage is the use of kernels or covariance functions to reduce the processing time. The best prediction performance was achieved with hybrid models developed by integrating GPR with RT models with the aid of dimensionality reduction and active learning techniques [56]. The majority of GPR-based mapping studies are aligned to use S2 data, which are used to retrieve multiple crop traits [50,78]. A few case studies are discussed in this section. Another interesting study is the quantitative estimation of soil organic carbon using S2 with the help of GPR [79]. Moreover, GPR models can be implemented into Google Earth Engine (GEE), which facilitates the large-scale spatio-temporal mapping of crop biophysical variables using Sentinel-3 (S3) data [80].

A Gaussian process is a combination of random variables d, which can be represented as in Equation (1), where $\mu$ expresses mean values and $\Sigma$ is the covariance matrix. It can further be derived as Equation (2).

$$X_i, \ldots, X_j \sim N(\mu, \Sigma) \tag{1}$$

$$p(x) = \frac{1}{\sqrt{(2\Pi)^d det|\Sigma|}} exp\left(-\frac{1}{2}(x-\mu)^T \Sigma^{-1}(x-\mu)\right), x = [x_{i,\ldots,}x_j]^T \in \mathbb{R}^d \tag{2}$$

A function can be sampled at point $x$, giving Equation (3), where $f(x)$ is the function, $m(x)$ is a mean function, and $k\left(x, x'\right)$ is a covariance function:

$$f(x) \sim GP(\left(m(x), k\left(x, x'\right)\right)) \tag{3}$$

If a training dataset and a test dataset contain N observations, they can be expressed as Equations (4) and (5):

$$D_{train} = (X, y) = \{x_i, y_i\}_{i=1}^N, \ x_i \in \mathbb{R}^d, y \in \mathbb{R} \tag{4}$$

$$D_{test} = X_* = \{X_{*,i}\}_{i=1}^{N'}, X_{*,i} \in \mathbb{R}^d \tag{5}$$

where $X$ is the matrix of training features, $X_*$ represents test points, and $y$ are training targets for regression.

The commonly used squared exponential kernel function ($k\left(x, x'\right)$) was selected for optimizing the GPR model and is expressed as

$$k\left(x, x'\right) = \exp(-\frac{\|x-x'\|^2}{2\sigma^2}) \tag{6}$$

where only the $\sigma$ hyperparameter needs to be adjusted.

The squared exponential is a widely accepted kernel function embedded in the GPR model for trait retrieval from both hyperspectral [11,81] and multispectral [80,82] datasets. The squared exponential kernel function is mostly used to reduce the total time taken for model training [81]. So, it is most commonly employed for retrieving crop traits from multispectral (e.g., [82]) and hyperspectral datasets (e.g., [57]). The MATLAB implementation of GPR with squared exponential kernel function was used in the present study for LAI and CCC retrieval.

### 2.4. Dimensionality Reduction Using Principal Component Analysis (PCA)

Because the acquired hyperspectral imagery consists of 269 bands, such a large number of contiguous bands may easily lead to suboptimal performance in the ML models due to spectral redundancy (Hughes phenomenon). To reduce redundancy and computational time while optimizing accuracy, a suitable dimensionality reduction strategy is to be applied. The MLRA toolbox in ARTMO provides eleven dimensionality reduction techniques for retrieving the most significant statistical variables. According to the analysis of all of them as a trade-off between accuracy and runtime in various settings, the PCA with 20 components is the most recommended one for retrieving multiple crop traits from hyperspectral datasets using GPR [54]. That strategy was also applied to the current analysis. In the case of PRISMA data used for retrieving multiple crop traits, GPR with PCA-20 achieved above 99.95% cumulative variance. Moreover, it slightly outperformed the band-ranking procedure with 20 bands [56].

### 2.5. Active Learning Methods and Field Verification

In RT modelling, a large, simulated dataset introduces redundancy and even makes it impossible to develop hybrid regression models for retrieving specific crop traits. In order to reduce the sample size without altering the model's predictive performance, noisy and reluctant samples may be removed by adopting an effective sample selection criterion. In solving regression problems related to the prediction of crop traits using Earth observation data products, two types of active learning (AL) methods are widely used, i.e., uncertainty and diversity methods [83,84]. The uncertainty approach includes pool active learning (PAL), entropy query by bagging (EQB), and residual active learning (RSAL). The diversity approach comprises Euclidean distance-based diversity (EBD), angle-based diversity (ABD), and clustering-based diversity (CBD) [85]. The MLRA toolbox in ARTMO facilitates applying these six AL methods as well as the random sampling techniques to the PROSAIL-simulated spectra in combination with a regression algorithm. Random sampling is the simplest approach to scaling down extensive datasets where there is no guarantee of reaching the most informative datasets compared with the original [86]. In the uncertainty approach, the samples were graded based on their uncertainties, and the least certain ones were taken as the output [86]. In the diversity approach, diversity refers to the variation among the samples considered, where the added samples show maximum dissimilarity with the ones already employed in the training dataset [87,88]. These AL methods select the most informative training samples for model development and prediction with the highest accuracy. In related studies, the EBD technique gave the best performance when integrated with GPR for retrieving LAI and LCC [85,89] and aboveground N content [84]. Also, the EBD-GPR models built using trait-specific AL-optimized training datasets from simulated S2 datasets facilitated the implementation of GPR models into Google Earth Engine (GEE) [90,91]. The efficacy of these AL techniques in retrieving various crop traits was well explained and analysed in [83–85].

The AL method selects samples from the pool dataset and trains the model. This takes place iteratively to check whether the data picked from the pool improve the model or not. Therefore, the samples showing no improvement are removed from the final dataset, and the process continues until all the samples in the pool dataset are evaluated against the validation dataset. Here, an in situ dataset of 45 samples and five non-vegetation spectra was used for validation purposes. The addition of non-vegetation spectra, usually soil spectra with respective trait values of zero, to the reduced sampling makes the model applicable to any UAV dataset with full heterogeneous classes [56]. Common goodness-of-fit statistics, such as mean absolute error (MAE), root mean square error (RMSE), normalized RMSE (NRMSE), and relative RMSE (RRMSE), are used for measuring retrieval accuracy. Altogether, five AL techniques, i.e., ABD, CBD, EBD, PAL, and RSAL, were exhaustively analysed and compared to find the optimal one for the proposed work.

## 3. Results

### 3.1. Model Evaluation and Selection of GPR

Eight multivariate models were evaluated for estimating LAI and CCC using PCA as a dimensionality reduction method with 20 components. The theoretical results showing the goodness-of-fit statistics calculated for each regression model are tabulated in Table 3. Validated against simulated data, GPR outperformed all other models in predicting the LAI by showing the highest $R^2$ value of 0.996, and KRR was found suitable for predicting the CCC with an $R^2$ value of 0.9997. The MAE values reported for LAI and CCC are 0.019 and 0.016, whereas the RMSE values are 0.143 and 0.024, respectively. In the case of NRMSE, the lowest values for estimating LAI and CCC are 1.946% and 0.433%, respectively. As evident from Table 3, based on NRMSE and $R^2$ values, the model performance sequence for LAI is GPR > KRR > NN > LS > ELM > BaTs > BoTs > SVR and that for CCC is KRR > GPR > NN > LS > ELM > SVR > BaTs > BoTs. Both GPR and KRR produced more accurate and robust results in estimating various crop traits. Since GPR possesses the unique characteristic of delivering uncertainties associated with mean estimates [51], the GPR algorithm was

selected for further training optimization applicable to UAV ultrahigh-spatial-resolution hyperspectral imagery for estimating crop traits.

**Table 3.** Accuracy assessment of MLRA models for retrieving LAI and CCC.

| S. No. | MLRA | MAE | RMSE | RRMSE (%) | NRMSE (%) | $R^2$ |
|--------|------|-----|------|-----------|-----------|-------|
| | | | LAI | | | |
| 1. | GPR | 0.019 | 0.143 | 3.796 | 1.946 | 0.996 |
| 2. | KRR | 0.114 | 0.153 | 4.065 | 2.084 | 0.995 |
| 3. | NN | 0.129 | 0.232 | 6.150 | 3.153 | 0.988 |
| 4. | LS | 0.189 | 0.247 | 6.545 | 3.355 | 0.987 |
| 5. | ELM | 0.189 | 0.321 | 8.499 | 4.357 | 0.978 |
| 6. | BaT | 0.218 | 0.357 | 9.469 | 4.854 | 0.976 |
| 7. | BoT | 0.303 | 0.413 | 10.933 | 5.604 | 0.963 |
| 8. | SVR | 0.334 | 0.449 | 11.892 | 6.096 | 0.957 |
| | | | CCC | | | |
| 1. | KRR | 0.016 | 0.024 | 1.562 | 0.433 | 0.9997 |
| 2. | GPR | 0.031 | 0.043 | 2.756 | 0.746 | 0.999 |
| 3. | NN | 0.031 | 0.050 | 3.262 | 0.904 | 0.999 |
| 4. | LS | 0.038 | 0.050 | 3.292 | 0.912 | 0.999 |
| 5. | ELM | 0.042 | 0.063 | 4.115 | 1.140 | 0.998 |
| 6. | SVR | 0.075 | 0.093 | 6.099 | 1.690 | 0.995 |
| 7. | BaT | 0.068 | 0.109 | 7.105 | 1.969 | 0.995 |
| 8. | BoT | 0.139 | 0.176 | 11.521 | 3.192 | 0.982 |

### 3.2. Performance of AL Techniques

Figure 3 represents the comparative results of various AL techniques for retrieving LAI and CCC. Smooth convergence can be noticed, which implies the usage of NRMSE over other statistical measures, such as $R^2$. The usage of NRMSE over $R^2$ for selecting the optimal AL was proposed in many earlier works [83,85]. The addition of a new sample at each iteration of the AL technique causes a stable decrease in RMSE and an increase in $R^2$ when validating with the field dataset. Optimal GPR model performance was achieved by a set of a few samples, i.e., 97 and 119 for LAI and CCC, respectively. On adding new samples, the NRMSE of RSAL decreases from 80.76 to 16.84 for LAI. In the case of CCC, the NRMSE of EBD shows a steady decrease from 21.6 to 19.42 for CCC. RSAL outperformed other AL techniques for LAI, while EBD showed the highest performance for CCC. The convergence observed at a low sampling size may be attributed to the low number of training data points used (*n* = 45).

### 3.3. Validation of Crop-Trait Models

The performance of AL-optimized GPR models for retrieving LAI and CCC was validated using in situ measurements from 45 plots. About five non-vegetation spectra (particularly, soil spectrum) were extracted from UAV hyperspectral imagery. Common statistical goodness-of-fit indicators, i.e., RMSE, NRMSE, MAE, and $R^2$, were used for validating the results. The validation results are shown as scatter plots between estimated and measured crop traits along with the goodness-of-fit statistics in Figure 4. Both LAI and CCC estimations show superior results with RMSE and MAE values less than 1. The RMSE values obtained for GPR models for LAI and CCC retrieval are 0.624 and 0.559, respectively. Further, the MAE values for LAI and CCC are 0.481 and 0.423, with $R^2$ values of 0.889 and

0.656, respectively. The most important statistical parameters, NRMSE values, reported for LAI and CCC retrieval are 8.579% and 14.842%, respectively.

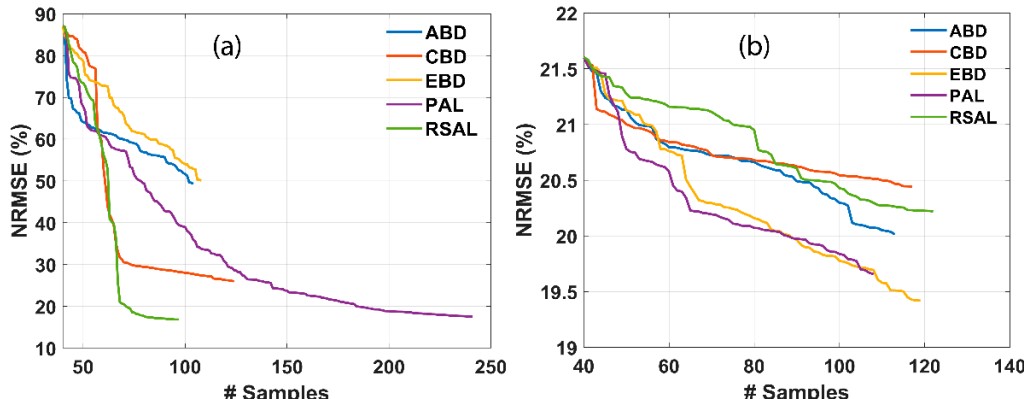

**Figure 3.** NRMSE (%) for several trait estimations using different AL methods. (**a**) LAI and (**b**) CCC. # samples denote the number of samples.

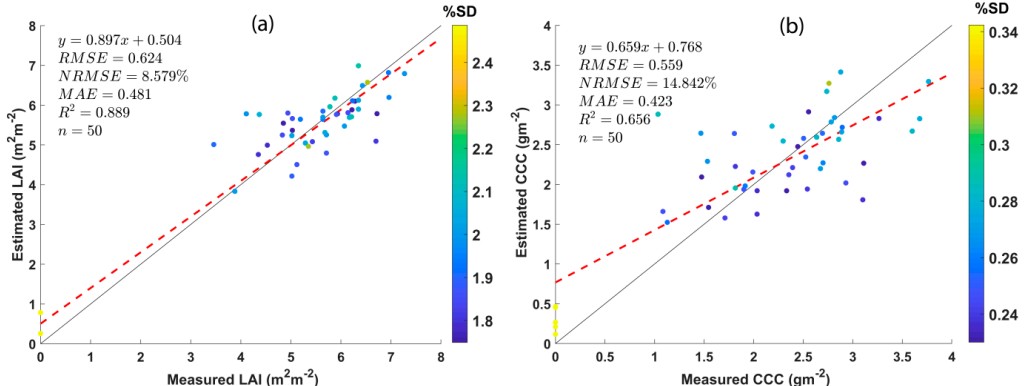

**Figure 4.** Scatter plots displaying the GPR model results against the in situ measurements along with goodness-of-fit statistics. (**a**) LAI and (**b**) CCC.

Another remarkable observation is the improvement in NRMSE thanks to the addition of non-vegetation spectra to the AL-optimized dataset and upon re-training. The NRMSE value for LAI was improved from 16.8 to 8.6%, and the CCC was lowered from 19.4 to 14.8%. Five bare-soil or non-vegetation spectra (10% of in situ measurements) were added to the AL-optimized dataset before the validation of crop traits.

### 3.4. Retrieval of LAI and CCC

The final GPR models were applied to pre-processed UAV hyperspectral imagery to obtain estimates and the accompanying uncertainties. The LAI and CCC retrieval maps and their associated coefficient of variation (CV) are shown in Figure 5. The in situ measurements for LAI and CCC range from 3.46 to 7.27 $m^2 m^{-2}$ and 1.03 to 3.76 g $m^{-2}$, respectively. On inspecting the retrieval maps, it is understood that they are strictly following the ranges of in situ measurements. The experimental plots with low values of LAI and CCC are clearly visible with red-coloured pixels, which indicate a strong pixel-wise variation of the retrieved values. For both the LAI and CCC maps produced, the maximum and minimum values appear on the same plots, which suggests that the retrieved maps are realistic and represent the best spatial variability. Even though the GPR models were trained with non-vegetation spectra added to the training with trait values set to zero, they were finally applied to a soil-masked pre-processed UAV image. Notably, no zero or close-to-zero values were obtained in the final estimated maps, which indicates the absence of non-vegetated regions, proving the advantage of including them during model training.

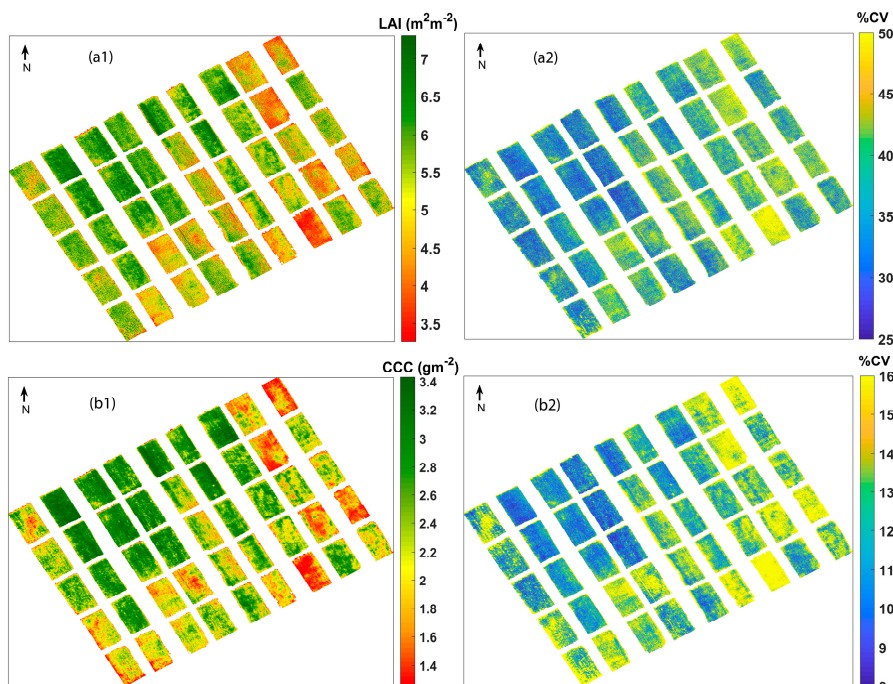

**Figure 5.** Mean estimates and coefficient of variation (CV) of (**a1**,**a2**) LAI and (**b1**,**b2**) CCC.

As a benefit of GPR, apart from estimates, associated uncertainties such as standard deviation (SD) and %CV maps on a per-pixel basis were provided for each parameter. Since SD is related to the magnitude of mean estimates, comparatively higher SD values were observed for chlorophyll content. To overcome this, a relative uncertainty measure, CV ((SD/mean estimate) × 100), expressed in % was used for the current analysis (Figure 5). It can be observed that high uncertainties were shown by pixels corresponding to lower values of crop traits. Usually, low values and values near zero (indicating bare soils) show absolute uncertainties of 1 or higher, while vegetation in good health and with proper irrigation shows higher values and low uncertainties [92]. So, these clear demarcations shown by uncertainty maps facilitate the spatial masking of regions with great certainty [48]. Moreover, it extends the portability of using the current model to other regions or acquisitions from other dates [51,93].

## 4. Discussion

Hybrid GPR models developed using PROSAIL simulations have already been successfully applied for retrieving crop traits from hyperspectral datasets from space-borne platforms [47,49]. Progressing along this line, this work attempts the retrieval of crop traits from UAV hyperspectral datasets at the ultrahigh spatial resolution of 4 cm. The retrieval accuracy and robustness of the models were validated using in situ measurements and showed the best results. During model evaluation, the top-most performance was shown by the nonlinear regression algorithms GPR, KRR, and NN, which are comparable with the results reported for multispectral S2 as well as hyperspectral datasets such as CHRIS and HyMap data in mapping LCC and LAI [48]. Both GPR and KRR are promising kernel-based regression algorithms that show stable and excellent results with the highest $R^2$ maxima. Even though KRR is mathematically simpler, with only one hyperparameter to optimize, the advantage of GPR is the usage of kernel functions and the delivery of predictive variance in addition to predictive mean [85]. The third best-evaluated model appeared to be NN; it mostly delivered poor results with a lack of robustness. Less training datasets and more noise tend to deliver unstable prediction results for NN, with a major problem of overfitting [48].

The advantage of using canopy RTMs for hybrid model training is that simulated spectra can be generated, satisfying ground situations, and these datasets can be optimized for the accurate retrieval of crop traits. Here, the crop traits of LAI and CCC of the wheat crops in IARI experimental fields were successfully retrieved using RT modelling applied to UAV hyperspectral imagery. This was achieved with the help of 2000 PROSAIL simulations, which are near to impossible to collect from the field on a particular date of UAV flight, along with field measurements of crop traits. The operational processing chains adopted for the study involve a hybrid approach integrating RT modelling with GPR. The application of a PCA with 20 components and optimum AL techniques with 97 and 119 samples for LAI and CCC, respectively, reduced the runtime and, at the same time, improved regression accuracy. Thus, it was proven that the integration of dimensionality reduction techniques and active learning methods leads to a significant decrease in the computation load involved in training GPR with large datasets [94,95]. The superior AL performance of EBD for retrieving LAI was also previously reported for multispectral [90] and hyperspectral [56] datasets. Likewise, other AL techniques implemented in ARTMO were recently also compared and selected based on the lowest NRMSE values for retrieving various crop traits, especially from hyperspectral datasets [57,96]. Likewise, here, we evaluated a few AL techniques, and it was found that RSAL and EBD led to the lowest NRMSE values and were thus selected for GPR prediction of LAI and CCC. Optimizing the training data is an important step in developing hybrid GPR models for retrieving crop traits from remotely recorded (usually noisy) spectral data. So, a list of AL techniques is applied against field data to select the most relevant representative simulations that match the real data [78]. This sampling reduction strategy reduces the large datasets of 2000 to manageable and more representative datasets, which considerably reduces the computational cost and guarantees the best stable and robust prediction results. The running of the AL techniques against in situ data was evaluated using NRMSE, because it provides information about the errors of the prediction model in estimating the response variables, whereas $R^2$ only provides information about correlation. The addition of non-vegetation spectra is an essential step in the validation part of GPR models and common practice in studies related to the retrieval of crop traits [53,55,56]. It helps the model to interpret non-vegetation regions by delivering results near zero. It is observed that the canopy variables of LAI and CCC provide low uncertainties, reaching close-to-zero estimates, as previously reported [56].

Previous studies reported higher inconsistencies in leaf-level traits for GPR-based retrievals [90,91], so only the canopy traits of LAI and CCC of wheat crops were estimated in the present study. The goodness-of-fit results obtained for the present study, especially the $R^2$ and NRMSE values of LAI and CCC, were in the acceptable range observed during the efficient retrieval of crop traits using other hyperspectral data [56]. The uncertainty maps help to evaluate the performance of regression models to predict values for regions other than training sites [51,93]. Since the associated uncertainty (SD) is related to the magnitude of estimates, chlorophyll shows increased SD values associated with the estimated trait [48]; therefore, the CV (SD/mean estimate, in %) was instead selected to examine the relative uncertainties. The lower uncertainty values obtained for LAI and CCC reveal more reliable retrieval values relative to the trained model. So, the variation in uncertainty estimates helps to evaluate the ground sampling strategy adopted for training the model as well as the portability of applying the model to images captured from new sites or at other times. All 45 plots of the wheat field were clearly discernible based on their field trait values and showed stable within-plot variability. A clear distinction of in situ values could be observed among plots and within plots, facilitating the use of these results for efficient farm nutrient management practices. Since these variables are clear indicators of plant stress, the retrieved within-field variability maps can be coupled with applied nutrient information to make decisions to improve crop health and yield [97].

While previous GPR studies delivered competitive results for hyperspectral datasets [53,89,98] by combining them with an appropriate dimensionality reduction technique [54], here,

an attempt was made to apply this approach to a hyperspectral UAV dataset with the ultrahigh spatial resolution of 4 cm for retrieving wheat crop traits. Crop traits in wheat were estimated using single-growth stage data. As a continuation of the present work, robust hybrid models will be built for estimating crop traits for a more diverse range of crop types acquired during multiple phenological stages and will be employed in the future using the same UAV sensor.

## 5. Conclusions

We introduced an integrated hybrid workflow for estimating wheat crop biophysical variables from unmanned aerial vehicle (UAV) hyperspectral imagery in the spectral range of 400–1000 nm. To develop hybrid models, the Gaussian process regression (GPR) machine learning regression algorithm was applied to PROSAIL simulations. Theoretical validation using eight regression models revealed the superiority of GPR. Suitable dimensionality reduction and active learning techniques were combined to mitigate the problems of redundancy and suboptimal model training. Two active learning (AL) techniques, i.e., residual active learning (RSAL) and Euclidean distance-based diversity (EBD), were selected for the GPR modelling of leaf area index (LAI) and canopy chlorophyll content (CCC). The low NRMSE values of 8.6% and 14.8% obtained for the GPR models during field verification suggest good retrieval accuracy and lower uncertainties for mapping LAI and CCC. Altogether, the proposed workflow offers the benefits of using a powerful (kernel-based) and generic hybrid approach for retrieving wheat crop biophysical variables from UAV datasets using ARTMO, a freely available software package. The developed workflow was successfully applied at the field level and can be upscaled for the quantitative and real-time mapping of vegetation products from farmers' fields using UAV technology.

**Author Contributions:** Conceptualization, R.N.S.; methodology, R.N.S., R.G.R. and J.V.; software, R.N.S. and J.V.; validation, S.G. and R.G.R.; formal analysis, S.G. and R.G.R.; investigation, R.G.R.; resources, R.R., T.K., M.C.M., J.M., A.D., S.K., M.K., R.D. and V.C.; data curation, R.R., T.K., M.C.M., J.M., A.D., S.K., M.K., R.D. and V.C.; writing—original draft preparation, R.G.R. and S.G.; writing—review and editing, R.N.S., S.G., R.G.R., J.V. and R.R.; supervision, R.N.S.; project administration, R.N.S.; funding acquisition, R.N.S. All authors have read and agreed to the published version of the manuscript.

**Funding:** The results summarised in the manuscript were achieved as a part of the research project "Network Program on Precision Agriculture (NePPA)", which is funded by Indian Council of Agricultural Research (ICAR), India, and is hereby duly acknowledged. J.V. was funded by the European Union (ERC, FLEXINEL, 101086622). The views and opinions expressed are, however, those of the author(s) only and do not necessarily reflect those of the European Union or the European Research Council. Neither the European Union nor the granting authority can be held responsible for them.

**Institutional Review Board Statement:** Not applicable.

**Informed Consent Statement:** Not applicable.

**Data Availability Statement:** Not applicable.

**Conflicts of Interest:** The authors report that there are no competing interests to declare.

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
