# Peer review of "Optimizing the Retrieval of Wheat Crop Traits from UAV-Borne Hyperspectral Image with Radiative Transfer Modelling Using Gaussian Process Regression"

_remotesensing, doi:10.3390/rs15235496_

Round 1
Reviewer 1 Report
This paper proposes a hybrid method combining a radiation transfer model with machine learning algorithms to extract leaf area and canopy chlorophyll content information from hyperspectral images obtained from unmanned aerial vehicle platforms. In field validation experiments, high accuracy was achieved. The article is well written and needed some revisions.
1. Accurately retrieving LAI can be achieved in some articles using the Beer-Lambert law. What is the reason for using PROSAIL?
2. Have the ill-conditioned problems in RTM inversion been considered?
3. Have all the evaluation indexes mentioned in the Active learning methods and Field verification section in lines 295-296 been used in subsequent sections? Are they used to evaluate the performance of the AL algorithm?
4. When there are multiple evaluation indexes, what was the ranking in line 310 based on?
5. Line 176, why is SMACC used? What are its advantages, and has it been compared with other methods?
6. Line 254, the library is about regression. What are the 11 specific dimensionality reduction techniques mentioned in the article?
7. Is it suitable to put lines 319-322 under the Result section?
8. The two sentences in lines 322-324 are redundant and can be combined into one sentence.
9. Lines 324-326, it is mentioned that according to the steady convergence shown in Fig. 3, the advantage of NRMSE over other statistical indexes can be seen. However, only NRMSE is shown in Fig. 3. If you want to compare its advantages with other evaluation indexes, please show the other evaluation indexes at the same time.
10. Line 326-327, it is mentioned that the AL technique of adding a new sample in each iteration leads to a stable decrease in RMSE. Is there any data to support this? This paragraph needs to be revised and the logic needs to be organized.
11. All the pictures in the article are not clear, and some symbols on the enlarged images cannot be seen clearly. The lower left corner of Fig. 1 is incomplete, and the legend of “( i )” is missing a small part on the top. Why is "#" added to the horizontal axis of Fig. 3?
12. Formulas in lines 230, 231, 234, 237, 238 are not numbered.
13. R2 should not be bolded in line 326.
14. The style of the tables in the text need to be unified.
The article is well written and needed some revisions.
Reviewer 2 Report
The manuscript examines the utility of hybrid modelling in retrieving LAI and CCC using UAV hyperspectral data. The results show that GPR outperformed other models that were tested in study. The results are exciting and very few studies have assessed the utility of hybrid models using remote sensing data captured using UAV. The standard of the paper is high and the English is very clear and concise. I thank the authors for putting together such piece of work for the vegetation remote sensing community.
General comments
However, I have a few questions that need responses from the authors.
· How did the authors match the scale scales at which LAI was measured and the spatial resolution of the hyperspectral imagery (4cm)?
· I understand that when comparing models there is need to bootstrap or perform a repeated cross validation on the calibration dataset (internal cross-validation) to avoid any trace of doubt on the results. Basing a comparison on a metric (R2 or RMSE) generated from a single run is never the best approach because these discrepancies observed might be due to chance. What I recommend is to bootstrap or cross-validate the model several times and generated a distribution of accuracy metrics (RMSE or NRMSE) and then compare the distributions of these metrics.
· The LUT generated from this study is too small – 2000 simulations, can the authors justify why such few simulations were generated. I also believe that before retrieving traits through hybrid model there is need to check how the simulations match the measured remote sensing data through a standard model inversion. This will demonstrate how well the RTM has managed to simulate the measured data before statistical methods are performed. Was noise added to the PROSAIL simulation?
Detailed comments
In the abstract report the NRMSE or RMSE instead of the R2. The coefficient of determination is not the best metric to assess retrieval accuracy.
Section 2.1- How many sample were collected in the field? There is need to provide information on how the sampling location were setup and how that sampling setup was harmonized with the remote sensing data captured using the UAV.
The authors need to add one or two sentences on how LCC and LAI was measured. How many above canopy measurements were measured Using the LI-COR LAI 2000 plant canopy analyzer. How the clumping effect was considered in the LAI measurements using the plant canopy analyzer.

Reviewer 3 Report
This manuscript retrieved wheat leaf area index and canopy chlorophyll content using Gaussian process regression in combination with different active learning algorithms. This paper has a clear structure, and is easy to understand. The main innovation is using GPR model for UAV-borne hyperspectral data with high accuracy of the two traits using in-situ measurements.
Major points:
1. As aforementioned, the innovation of this study mainly resides in using a UAV dataset and optimizing the training process by sing active learning methods. However, as mentioned in the method section, there have already been several studies on integrating GPR with active learning techniques such as references 82 and 86. My question thus would be what is the added value of this current paper?
2. The introduction is too long and could be more concise in my point of view.
3. Section 2.5, what do you mean by “the uncertainty approach” and “the diversity approach”? Is there any reference on these terms? Line 277, why are the samples with least certainty be taken as output?
Minor points:
1. Line 21, “A hyperspectral image captured from…”, was it that only one UAV image was required or used in this study?
2. Line 43, and strongly associated with -> and is strongly associated with
3. Line 44, one of key factors -> one of the key factors
4. Line 50, measures -> measure
5. Line 82, I think this sentence is grammatically flawed, please check.
6. Line 312-313, please give a reference of this characteristic of GPR and this information could be given in the method section.
7. Line 458, the present study only used on dimension reduction method, which is PCA, without validation in saying that it is the “suitable” technique. Also, it would be better to avoid abbreviations in the conclusion section.
Only minor editing of English language is required.
Round 2
Reviewer 3 Report
The authors have addressed many questions raised. I have only one major suggestion. As pointed out in the introduction section, one unique contribution of this study is optimization of the hybrid GPR model using AL sampling methods. However, this is totally absent from the current abstract. Thus it is advised to be added. Based on this concern, the title of this study could be revised to highlight its distinguishing features, maybe also adding the word "optimized" in it.
A minor reminding is to pay attention to the redundant or lack of "space" in the manuscript, e.g. Line 141, Line 97.
